# L-tyrosine-bound ThiH structure reveals C–C bond break differences within radical SAM aromatic amino acid lyases

Patricia Amara ⬢ [1], Claire Saragaglia[1], Jean-Marie Mouesca ⬢ [2], Lydie Martin[1] & Yvain Nicolet ⬢ [1✉]

2-iminoacetate synthase ThiH is a radical *S*-adenosyl-L-methionine (SAM) L-tyrosine lyase and catalyzes the L-tyrosine Cα–Cβ bond break to produce dehydroglycine and *p*-cresol while the radical SAM L-tryptophan lyase NosL cleaves the L-tryptophan Cα–C bond to produce 3-methylindole-2-carboxylic acid. It has been difficult to understand the features that condition one C–C bond break over the other one because the two enzymes display significant primary structure similarities and presumably similar substrate-binding modes. Here, we report the crystal structure of L-tyrosine bound ThiH from *Thermosinus carboxydivorans* revealing an unusual protonation state of L-tyrosine upon binding. Structural comparison of ThiH with NosL and computational studies of the respective reactions they catalyze show that substrate activation is eased by tunneling effect and that subtle structural changes between the two enzymes affect, in particular, the hydrogen-atom abstraction by the 5´-deoxyadenosyl radical species, driving the difference in reaction specificity.

[1] Univ. Grenoble Alpes, CEA, CNRS, IBS, Metalloproteins Unit, F-38000 Grenoble, France. [2] Univ. Grenoble Alpes, CEA, CNRS, IRIG-DIESE-SyMMES-CAMPE, 38000 Grenoble, France. ✉email: yvain.nicolet@ibs.fr

Thiamine pyrophosphate (TPP), derived from vitamin B1, is a key ubiquitous cofactor[1] for many enzymes involved in carbohydrate metabolism[2]. It enables, for example, the formation of acetyl coenzyme A by pyruvate dehydrogenase[3] and pyruvate: ferredoxin oxidoreductase[4] as well as the transfer of a glycoaldehyde from a ketosugar to an aldosugar by transketolase[5]. Dehydroglycine (DHG), also termed 2-iminoacetate, is a key precursor in the bacterial biosynthesis of the thiazole ring that constitutes the core of TPP. In aerobic prokaryotes, such as the extensively studied *Bacillus subtilis*, DHG is produced by the oxygen-dependent glycine oxidase[6,7]. Such glycine oxidation is not possible under anaerobic conditions and anaerobes, including the facultative bacterium *Escherichia coli*, use instead the 2-iminoacetate synthase ThiH that catalyzes DHG formation from L-tyrosine processing (Fig. 1)[8–11].

ThiH is a member of the radical *S*-adenosyl-L-methionine (SAM) superfamily[12,13] and contains one [Fe₄S₄] cluster that reductively cleaves SAM to form both methionine and a highly reactive 5′-deoxyadenosyl radical (5′-dA•) species. Subsequently, 5′-dA• abstracts from L-tyrosine, a hydrogen atom, long proposed to be the phenolic hydrogen atom[9,10]. By analogy, the same reaction step would be catalyzed by the radical SAM tyrosine lyase HydG involved in the assembly of the [FeFe]-hydrogenase active site[14,15]. ThiH and HydG are closely related members of the radical SAM superfamily sharing 27% sequence identity[16]. Both enzymes catalyze the cleavage of L-tyrosine Cα–Cβ bond to generate *p*-cresol[10,16] and DHG[9,17]. The latter is used by HydG at a second active site of the enzyme to synthesize the CO and CN⁻ diatomic ligands[18–20] that are part of the so-called [FeFe]-hydrogenase H-cluster[21,22]. Conversely, ThiH transfers DHG to ThiG, where it is further processed to produce the thiazole ring of thiamine[11].

These two tyrosine lyases are often compared to the radical SAM tryptophan lyase NosL[23], involved in the synthesis of the thiopeptide antibiotic nosiheptide[24]. NosL shares about 23% primary structure identity with them[25]. But, whereas ThiH and HydG cleave the L-tyrosine Cα–Cβ bond[9,10,16,17], NosL cleaves the L-tryptophan Cα-C bond instead and produces a 3-methylindole-2-carboxylic acid (MIA)[26,27] (Supplementary Fig. 1). Unexpectedly, the crystal structure of L-tryptophan-bound NosL[25] showed that 5′-dA• is well positioned to abstract the L-tryptophan amino hydrogen atom, yielding a NH• tryptophanyl radical (called [L-Trp-NH•] herein), an observation supported by a subsequent biochemical study[28]. In addition, structure-based sequence comparisons of NosL with tyrosine lyases ThiH and HydG show that, in the latter enzymes, the corresponding [L-Tyr-NH•] would be formed[25]. Using electron paramagnetic resonance (EPR), a methylene-centered *p*-cresyl radical intermediate (*p*-cresyl•) was trapped during L-tyrosine scission by HydG[15] and, more recently, the 5′-dA• was trapped using *p*-coumaric acid instead of L-tyrosine[29].

Here, we report the crystal structure of ThiH from *Thermosinus carboxydivorans* (*Tc*) in complex with its L-tyrosine substrate. In this study, an unexpected substrate protonation state, supported by its interactions in our X-ray model, has been confirmed by molecular dynamics (MD) simulations. More specifically, deprotonation of the L-tyrosine hydroxyl moiety is in agreement with the very recent report that, in HydG, the ensuing *p*-cresyl• intermediate is most probably deprotonated as well, and corresponds to a 4-oxidobenzyl radical instead[30]. The description of the mechanism of direct hydrogen atom abstraction by 5′-dA• at the amino nitrogen position of L-tyrosine, obtained by using hybrid quantum mechanical / molecular mechanical (QM/MM) methods, supports a highly controlled and tight local environment, inducing a tunneling effect during hydrogen transfer that we have estimated analytically to be about 50% of the reaction barrier. Additional calculations performed on both ThiH and NosL show that this step corresponding to the radical formation [L-Tyr-NH•] ([L-Trp-NH•] in NosL) is crucial for the orientation of the subsequent bond cleavage step (i.e. Cα–Cβ and Cα-C in ThiH and NosL, respectively). In addition, comparison of ThiH with HydG and NosL has revealed unique structural motions, most likely responsible for specific substrate access and product release sites, showing that these enzymes work as assembly lines. Thus, by combining X-ray crystallography and computational chemistry, we provide a rarely achieved mechanistic insight into the hydrogen-atom abstraction step catalyzed by radical SAM enzymes and underscore how subtle changes at the active site afford both substrate selectivity and chemical regiospecificity.

## Results

**TcThiH overall structure**. *Tc*ThiH was incubated with its substrate L-tyrosine, 5′-deoxyadenosine (5′-dA) and methionine as mimics of the SAM cleavage products prior to crystallization. Brownish prism-shaped crystals that diffracted to 1.27 Å resolution were obtained. In this crystal form, two independent ThiH molecules termed chains A and B were present in the asymmetric unit. Surface interaction calculations using the PDBePISA server[31] supported the fact that the protein does not form quaternary structures in the crystals and would likely correspond to monomers in solution, in agreement with our previous observation when performing size exclusion chromatography. The refined model (see Methods and Supplementary Table 1) contains residues 2–367 (molecule A) and 1–367 (molecule B) with a root mean square deviation (RMSD) of 0.3 Å over 265 Cα atoms between the two molecules. The radical SAM domain includes residues 71 to 356

**Fig. 1 Reaction catalyzed by ThiH in thiamine biosynthesis.** Scheme of the reductive cleavage of *S*-adenosyl-L-methionine into L-methionine and the 5′-deoxyadenosyl radical. The latter abstracts a hydrogen atom from L-tyrosine to form *p*-cresyl• and dehydroglycine.

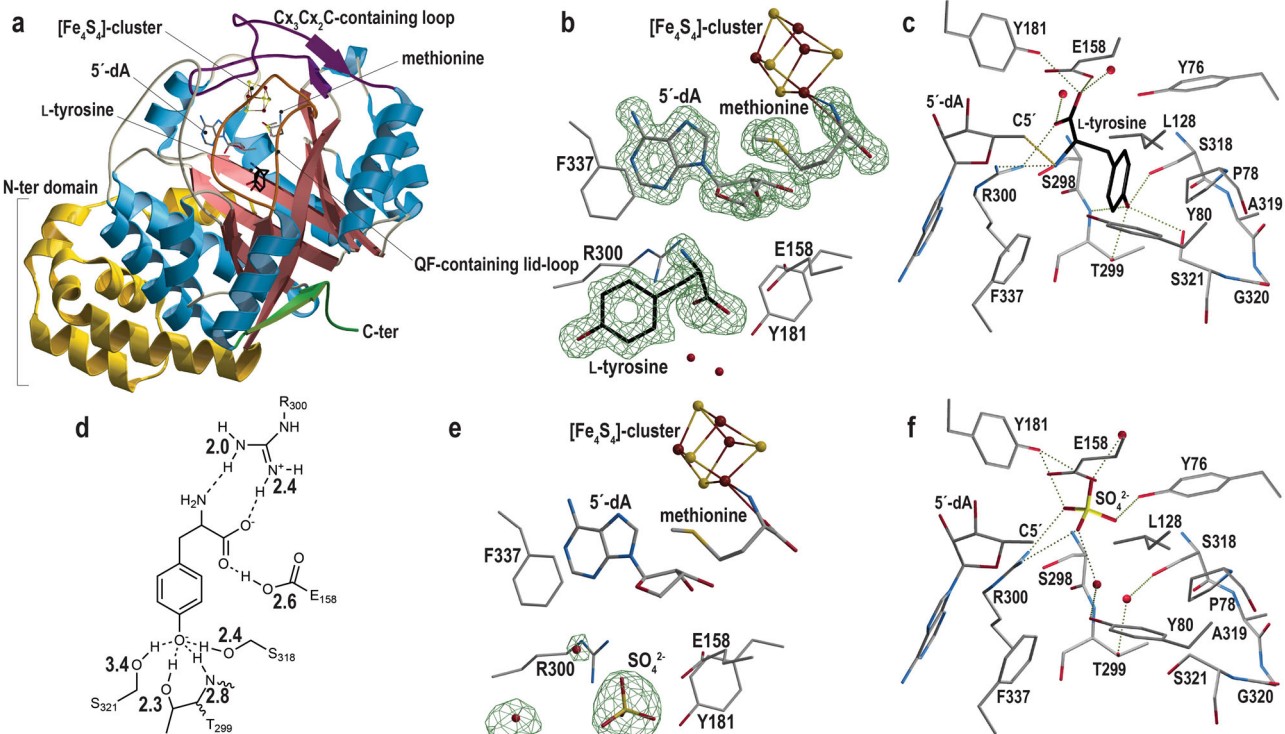

**Fig. 2 Crystal structure of *Tc*ThiH. a** Overall structure of ThiH where the radical SAM core is depicted in indian red (β-strands) and blue (α-helices). The N- and C-terminal stretches are depicted in gold and green, respectively. The CX₃CX₂C-motif containing loop and the additional β-sheet are represented in purple. The [Fe₄S₄]-cluster is depicted in ball-and-stick while 5′-dA, the L-methionine bound to the [Fe₄S₄] cluster and L-tyrosine are represented as sticks. C, N, O, S and Fe atoms are represented in light gray (dark gray for L-tyrosine), blue, red, yellow and brown, respectively. **b** Zoom of the L-tyrosine *Tc*ThiH binding site. The $F_o − F_c$ difference Fourier (omit) electron density map for 5′-dA, L-methionine and L-tyrosine is contoured at 4.5 σ and depicted as green mesh (cover radius 2.0 Å). Atoms are represented and colored as in (**a**). **c** Substrate (L-tyrosine) binding mode at the *Tc*ThiH active site. Atoms are represented and colored as in (**a**). The $2F_o − F_c$ electron density map of L-tyrosine, that is not shown here to visualize the substrate's interactions with the protein, can be found in Supplementary Fig. 4a. **d** Scheme of interactions between the L-tyrosine and the *Tc*ThiH active site; Distances in Å (X-ray model, molecule A), between the heavy atoms of the protein and those of the substrate, are indicated at the protein atoms for clarity. The atomic coordinate errors have been estimated to 0.036 Å for all the L-tyrosine heavy atoms (0.046 Å for the overall structure) using the Online_DPI server[56]. **e** Zoom of the substrate-free *Tc*ThiH active site in the same orientation as in (**b**) (see main text). The $F_o − F_c$ difference Fourier (omit) electron density map surrounding the sulfate ion and the water molecules is contoured at 3 σ and depicted as green mesh (cover radius 2.0 Å). **f** Substrate-free *Tc*ThiH active site in the same orientation as in (**c**). Stereoviews of (**b**, **c**, **e**, **f**) are depicted in Supplementary Figs. 4b, 5, 6 and 7, respectively.

and corresponds to a full (βα)₈ triose-phosphate isomerase-like fold flanked by N- and C-terminal stretches (Fig. 2a). Residues 1–71 define a compact α-domain (in gold in Fig. 2a) similar to that observed in radical SAM proteins BioB[32], HydE[33], PylB[34], HydG[35,36] and NosL[25] (Supplementary Fig. 2). The short C-terminal stretch starts at residue G356, which belongs to a semi-conserved GxxP motif that defines the end of the radical SAM β-barrel in the above-mentioned enzymes. Strikingly, when comparing molecules A and B, this stretch displays different conformations, highlighting structural flexibility in this region. For instance, in molecule B, it corresponds to a short β-strand that interacts with the N-terminal part of first β-strand of the radical SAM core domain, followed by a coil that covers the barrel and closes the cavity. By contrast, in molecule A, due to crystal-packing contacts with molecule B, this segment is pushed away and folds as a short α-helix, hence leaving the barrel cavity accessible to solvent (Supplementary Fig. 3). Inspection of the electron density map indicates features that can be best modeled as 5′-dA, L-methionine bound to the radical SAM [Fe₄S₄]-cluster and L-tyrosine (Fig. 2b). The latter is embedded in the active-site cavity in an orientation reminiscent to that observed in NosL-bound L-tryptophan.

**L-tyrosine binding mode to the *Tc*ThiH active site.** The amino nitrogen atom interacts with Nη2 of the strictly conserved residue

R300 and points toward the C5′-atom of 5′-dA, supporting a direct hydrogen-atom abstraction from that position by the 5′-dA• species. The carboxyl moiety is sandwiched between the strictly conserved residue R300 Nη1- and E158 Oε1-atoms (Fig. 2d). These unexpected interactions suggest an unusual protonation state to avoid electrostatic repulsions between the amino and guanidinium groups on the one hand and the two carboxyl moieties on the other hand. The phenol ring sits in a narrow hydrophobic pocket while the phenol oxygen atom establishes four polar interactions with the hydroxyl groups of the strictly conserved residues T299, S318 and S321, in addition to the main-chain nitrogen amide of residue T299, hence defining a square pyramidal geometry (Fig. 2c) supporting anion-dipole interactions instead of hydrogen-bonding. In order to investigate the most probable protonation state, we performed QM/MM geometry-optimizations of L-tyrosine-bound *Tc*ThiH using different combinations of protonation states for L-tyrosine and residues R300 and E158 (see Supplementary Methods). We have also performed a 250-ns molecular dynamics (MD) simulation of L-Tyr bound *Tc*ThiH (Supplementary Methods and Supplementary Fig. 8) at the optimal bacterial growth temperature (60 °C). Both approaches show that only the phenolate-containing models were able to reproduce the phenol oxygen atom interactions with the protein observed in the crystal structure, hence supporting deprotonation of the phenol upon

L-tyrosine binding (Fig. 2d). In addition, the strictly conserved residue R306 located downstream the T299, S318 and S321 forming crown, is pointing toward the phenolate further stabilizing this deprotonated state. Noteworthy, when comparing the ThiH structure with that of the tyrosine lyase HydG, the corresponding T299, S318, S321 and R306 in HydG are strictly conserved, highlighting their importance in the specific recognition and binding of L-tyrosine in both enzymes. Regarding the carboxyl moiety, QM/MM calculations support a deprotonated state interacting with a protonated E158 carboxyl group. In striking contrast, E158 corresponds to an asparagine in HydG. Yet, a strictly conserved glutamate residue is observed at position 128, hence compensating for this E to N substitution. A direct carboxylic acid carboxylate interaction may be important to favor the Cα–Cβ bond break over alternative radical-based β-scission. When considering the interaction between the L-tyrosine amino group and residue R300, QM/MM geometry-optimizations suggest a L-tyrosine with a deprotonated $NH_2$, facing the R300 guanidinium as was already proposed for NosL-bound L-tryptophan[25,37]. Therefore, upon binding, L-tyrosine would be fully deprotonated, likely corresponding to $NH_2$-CH-(Ph-$O^-$)($COO^-$) (Fig. 2c, d).

**Structural comparison of ThiH with HydG.** A close primary structure comparison between tyrosine-lyases ThiH and HydG indicates that, aside from the E158 to N substitution compensated by a glutamate/aspartate residue in HydG at position 128 (*Tc*ThiH numbering), all the key residues involved in L-tyrosine binding are strictly conserved in both proteins (Fig. 3a) supporting similar L-tyrosine interactions in both enzymes. Yet, previous structural characterizations of HydG failed in obtaining L-tyrosine-bound structures[35,36]. As a consequence, the corresponding active site cavity in HydG crystal structures display significant structural differences with ThiH concentrated on both the R300-containing loop and the S318-N336 segment (Supplementary Fig. 9). These rearrangements were proposed to be necessary for substrate access and suggested an induced-fit upon L-tyrosine binding[36]. In order to determine whether the ThiH active site cavity would follow the same trend in the absence of substrate, *Tc*ThiH was incubated only with 5′-dA and methionine prior to crystallization. The as-obtained crystals diffracted to 2.0 Å and belonged to the same space group as previously (Supplementary Table 1). The overall structure is identical to that of the L-tyrosine-bound *Tc*ThiH with a RMSD of 0.19 Å (molecule A) and 0.25 Å (molecule B), in the same range derived from comparing molecules A and B of the same crystal (see above). The *Fo-Fc* residual electron density map at the substrate-binding site indicates features that do not correspond to L-tyrosine, best modeled as two water molecules and a sulfate ion from the crystallization condition (Fig. 2e). Hence, the absence of substrate did not lead to the collapse of the active site architecture, at odds with what was observed for HydG[35,36]. Strikingly, all the corresponding side-chain residues remained in the same conformation. The sulfate ion sits at the same position as that of the L-tyrosine carboxyl moiety, facing residue E158 further supporting protonation of this residue even in the absence of substrate. Conversely, in the absence of anion at the phenolate binding site, there is a ~110° rotation of the S321 residue side-chain compared to its location when the substrate is bound. The water molecule at that position establishes only two hydrogen bonds (Fig. 2f). During the MD simulations of L-tyrosine bound ThiH, the rotation of S321 is occasionally observed (Supplementary Figs. 8b, 10).

**Structural comparison of ThiH with NosL.** We next compared the L-tyrosine-bound *Tc*ThiH structure with the one of the previously reported L-tryptophan-bound NosL from *Streptomyces actuosus* (*Sa*)[25], in order to identify what makes such difference in

substrate specificity. *Tc*ThiH and *Sa*NosL share about 23% identity mainly located at key positions either to stabilize the tridimensional architecture or because they are important for their function. Yet, the two structures are very similar and display a RMSD of 2.1 Å for 281 Cα-atoms superimposed. The main structural differences lay in the N-terminal α-domain. As can be seen in Fig. 3b the L-tyrosine and L-tryptophan binding sites are very similar and most of the residues surrounding the two substrates are either conserved or equivalent in both structures. Likewise, L-tyrosine and L-tryptophan orientations are identical. The amino nitrogen is facing the 5′-dA C5′-atom and a strictly conserved arginine (R300 and R323 in *Tc*ThiH and *Sa*NosL, respectively). Their respective dihedral angles are also the same. Strikingly, the main significant difference corresponds to a one-residue insertion at the C-terminal end of strand β8 (Fig. 3a, b) in ThiH, thereby generating a β-bulge, which ideally places residue S321 to establish the fourth polar interaction with the L-tyrosine phenolate. The latter is further stabilized by an arginine residue (R306 in ThiH) located right beneath the β-bulge and conserved in tyrosine lyases ThiH and HydG, but absent in NosL. Hence, the combination of the one-residue insertion and this basic residue are likely to be the primary markers for selecting L-tyrosine versus L-tryptophan. Yet, these differences are located at the tip of the aromatic rings and cannot explain why tyrosine lyases and NosL selectively cleave the Cα–Cβ and Cα–C bond, respectively. The only notable difference between the two enzymes in the vicinity of the Cα-atom is the presence of a direct interaction between the L-tyrosine carboxyl moiety and E158 in ThiH, whereas the L-tryptophan carboxyl is surrounded only by water molecules in NosL.

The structural information we obtained on the L-tyrosine binding mode at the *Tc*ThiH active site (Fig. 2c, d) prompted us to perform QM/MM calculations (see Supporting Methods for details) to further investigate its reaction mechanism (Fig. 1) but also to explain the different regioselectivity of NosL which, despite its sequence similarity with ThiH and HydG and a structurally close active site, catalyzes instead the scission of the Cα–C bond of L-tryptophan (Supplementary Fig. 1).

**Tunneling effect during hydrogen-atom abstraction by 5′-dA•.** Starting from the L-tyrosine bound *Tc*ThiH structure (this work) and the L-tryptophan bound *Sa*NosL structure we had previously solved[25], both crystallized with 5′-dA, we first focused on the 5′-dA• attack on their respective substrates. Since the attack occurs for both models at the substrates' amino group (Supplementary Fig. 13), we modeled 5′-dA• and scanned the substrate amino group position by geometry-optimizing the whole protein using QM/MM potentials (Supplementary Methods) while holding the corresponding dihedral angle Φ (Cβ-Cα–N-H1 of L-tyrosine or L-tryptophan, see Fig. 4 and Supplementary Figs. 13, 14a, b). Even though it is difficult to point out which of the ThiH and NosL active site structural features may orient the reaction differently (Fig. 3a, b), this series of calculations reveals one global minimum with Φ = −47.8° and one global and one reachable minima (Φ = −47.9° and Φ = 55°) for [5′-dA• + L-Tyr]-*Tc*ThiH and [5′-dA• + L-Trp]-*Sa*NosL, respectively (Supplementary Fig. 14c). Thus, regarding substrate orientations in their respective active sites, the global minima of both enzymes are very similar along the lines of their structural similarity (see previous section). Interestingly, an orbital analysis of the first five points on the dihedral scan for both enzymes (Supplementary Fig. 15) shows that, for ThiH, the lowest unoccupied molecular orbital (LUMO) persistently remains located on the C5′ moiety whereas the LUMO + 1 partner is located on the adenine part. Both orbitals remain well separated by an average energy gap of ~0.6 eV. This stability in the orbital characters, relative orders and

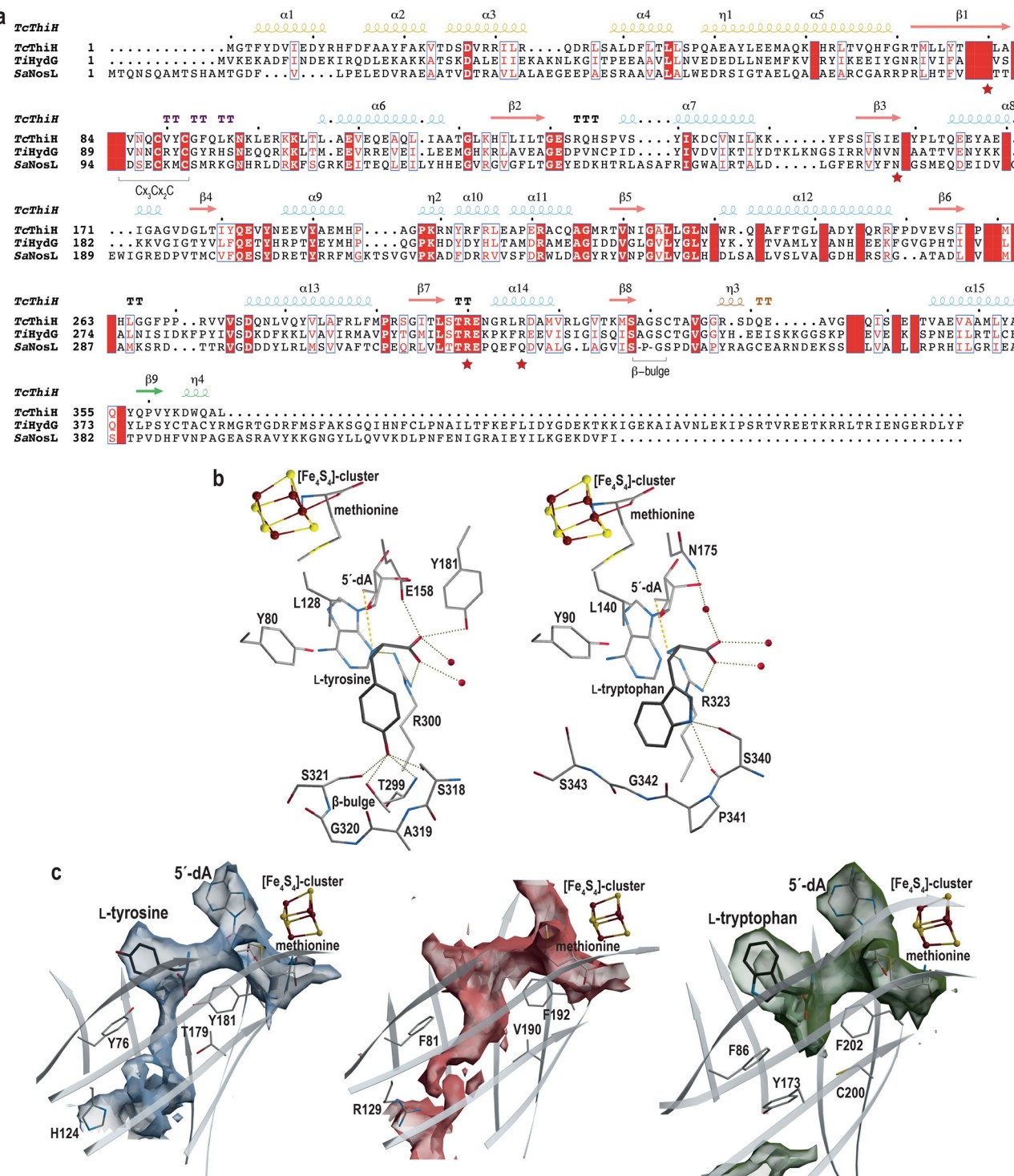

**Fig. 3 Comparison *Tc*ThiH with *Ti*HydG and *Sa*NosL. a** Multiple sequence alignment of *Tc*ThiH, HydG from *Thermoanaerobacter italicus* (*Ti*) and *Sa*NosL primary structures. Red and white boxes indicate conserved and similar amino acids in the three proteins. *Tc*ThiH secondary structure elements are depicted on top following the same color code as in Fig. 2a. Red stars highlight residues 80, 158, 300 and 306 that are discussed in the text. **b** L-tyrosine (left) and L-tryptophan (right) binding mode in *Tc*ThiH (this work) and *Sa*NosL[25], respectively. Except for the [Fe₄S₄] cluster depicted in the ball-and-stick form, the rest of the active site and the substrate are represented as sticks and water molecules are indicated as red spheres. C, N, O, S and Fe atoms are represented in light gray (dark gray for the substrates), blue, red, yellow and brown, respectively. **c** The cavity maps, contoured with an accessible probe radius of 1 Å, were calculated using the X-ray models of *Tc*ThiH (left, steel blue), *Ti*HydG[35] (center, ruby-red) and *Sa*NosL (right, olive color). When present in the structural model, 5′-dA, the L-methionine bound to the [Fe₄S₄] cluster and the substrate were removed for the cavity calculation. Key residues and the [Fe₄S₄] are represented as in (**b**). Stereoviews of (**b**, **c**), are presented in Supplementary Figs. 11, 12, respectively.

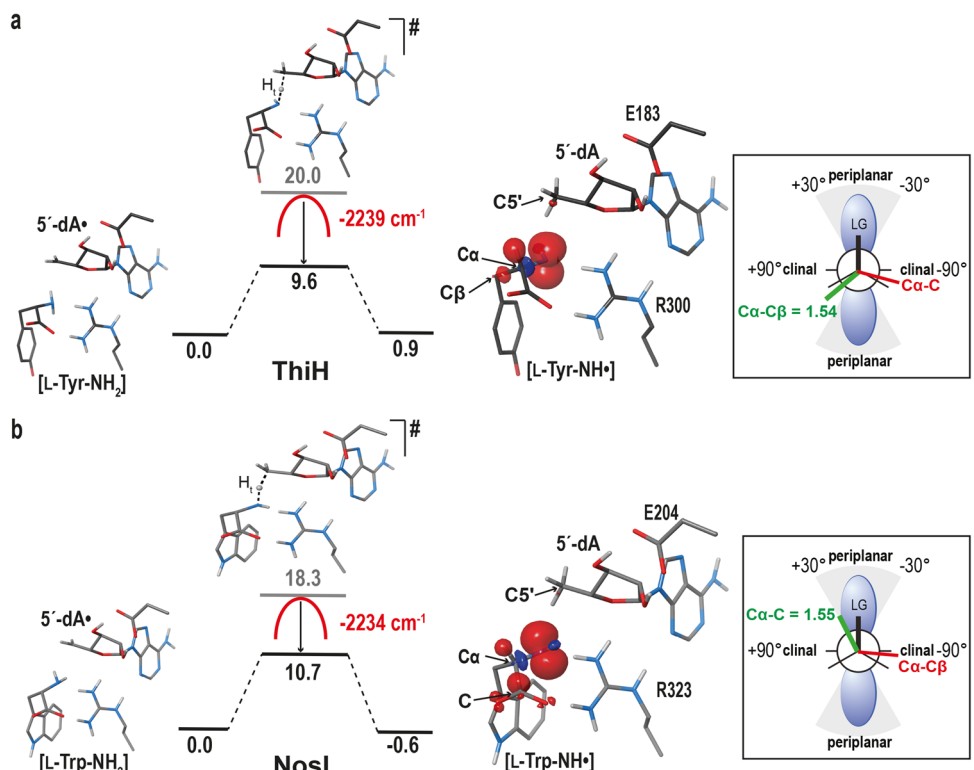

**Fig. 4 Substrate hydrogen abstraction by 5´-dA•.** **a** Reaction energy profile for the attack by 5´-dA• leading to the formation of [L-Tyr-NH•] - $\Phi$ (Cβ-Cα–N-H) = 39.6°, ThiH product P in Supplementary Table 2 - within the *Tc*ThiH active site. Insert (right): Newman representation, adapted from reference[57], of the dihedral angle between the partially occupied p orbital (blue) at the nitrogen radical center and the Cα–Cβ bond (in green) or the Cα–C bond (in red), where *p*-cresyl• (or •COO⁻) is the leaving group (LG). The periplanar region is indicated in gray, the rest being clinal. Although not optimal, the orbital overlap allows the Cα–Cβ bond break while the Cα–C bond break is not possible with the dihedral angle clearly lying in the clinal region. **b** [L-Trp-NH•] - $\Phi$ (Cβ-Cα-N-H) = 174.5°, NosL product P in Supplementary Table 2 - obtained from the NosL *productive* reactant R - $\Phi$ (Cβ-Cα-N-H1) = 55°, Supplementary Fig. 14c - within the *Sa*NosL active site. Insert (right): Newman representation as in (**a**), showing that, in this case, the orbital overlap is maximized to break the Cα–C bond with the corresponding dihedral lying in the periplanar region. By contrast, the dihedral corresponding to the Cα–Cβ bond break clearly lies in the clinal region, impeding this reaction to occur. In (**a**, **b**), single point energies for reactants and products are given in kcal.mol⁻¹; the TS energy in gray corresponds to the calculated value from the IRC calculations while the one in black is the barrier taking into account the tunneling effect (see main text and Supplementary Methods for details); the value of the TS single imaginary frequency is indicated in red; for the reactant, transition state and product, a minimal model of the active site extracted from the corresponding QM/MM geometry is represented as sticks with C, N, O and H atoms in gray, blue, red and light gray, respectively; except for the 5´-dA C5´ atom, only polar hydrogens are shown for clarity; the hydrogen atom that is transferred is represented as a light-gray ball in the transition state; the spin density is shown for the product (the product geometries without the spin density are shown in Supplementary Fig. 17).

energies is consistent with the steady energy increase observed in Supplementary Fig. 14a for the first five points of the scan, leaving only one minimum ($\Phi = -47.8°$) at the start of the scan. By contrast, for NosL, the C5´ and adenine parts are energetically crossing, i.e. the characters of LUMO and LUMO + 1 are exchanged and their corresponding energies remain close within ~0.2 eV range. This means that 5´-dA• is influenced by subtle changes in its environment. This results in an inflection of the energy increase (Supplementary Fig. 14b), generating a secondary local minimum ($\Phi = 55°$) in addition to the first lowest point ($\Phi = -47.9°$) mirroring that of ThiH. As it turns out, this energy scan inflection for NosL opens an alternative path leading ultimately to the Cα–C bond scission whereas the absolute minimum common to both ThiH and NosL would lead for NosL to the nonproductive Cα–Cβ bond scission (Supplementary Fig. 16). In spite of our efforts, we could not pinpoint the precise origin of this difference in the interaction of 5´-dA• with their respective environments because, as already mentioned above, structural comparison shows that the L-tyrosine and L-tryptophan binding sites are very similar with most surrounding residues being either conserved or equivalent (Fig. 3b). Yet, the sum of these very

minute structural differences leads to slightly different electronic structures of their active sites.

We next investigated the hydrogen abstraction by 5´-dA• starting from the models we obtained for [5´-dA• + L-Tyr]-*Tc*ThiH and [5´-dA• + L-Trp]-*Sa*NosL. Scanning the position of the hydrogen atom abstracted by 5´-dA• (step 2 in Fig. 1), we calculated the transition state leading to [5´-dA + L-Tyr-NH•]-*Tc*ThiH. In this state, we observed that the Cα–Cβ bond is weakened by an electron depletion (see spin density in Fig. 4a). Using the same procedure, we obtained the transition state for [5´-dA + L-Trp-NH•]-*Sa*NosL where, this time, the Cα–C bond is depleted. The latter model (Fig. 4b) corresponds to the reactant local minimum at $\Phi = 55°$ (Supplementary Fig. 14b, c); we found that the global minimum of [5´-dA• + L-Trp]-*Sa*NosL at $\Phi = -47.9°$ weakens the Cα–Cβ bond instead when the hydrogen atom is transferred to 5´-dA• (see Supplementary Fig. 16, Supplementary Table 2 and discussion below).

The transition state geometries for these reactions were validated by frequency calculations. Strikingly, the unique imaginary frequency exhibits a large absolute value greater than 2000 cm⁻¹, supporting the occurrence of a tunneling effect (Fig. 4 and

Supplementary Fig. 16; see also Supplementary Table 3)[38]. In addition, the calculated reaction barriers for the productive hydrogen-atom abstraction in ThiH and NosL were high (20.0 and 18.3 kcal.mol$^{-1}$, respectively). Hence, we decided to further investigate the impact of a hydrogen tunneling effect for the reaction to proceed (Supplementary Table 4). As it turns out, upon formation, the highly reactive 5′-dA• radical is tightly embedded in the substrate-filled cavity nearby the targeted substrate hydrogen atom to abstract. In addition, the large absolute frequency values we found for both transition state geometries imply very short transfer distances; we find these to be less than 0.5 Å, from a set of five QM/MM geometries from the intrinsic reaction coordinate (IRC) path (Supplementary Methods and Supplementary Table 5). Therefore, this suggests, in both ThiH and NosL cases, that the local short and tight movement induced by the hydrogen transfer from the substrate amino group NH$_2$ to the C5′ atom of 5′-dA• radical is decoupled from the otherwise *slower* movements of the rest of the enzyme[39–42]. Thus, to evaluate the energy gain obtained by the tunnel effect, we used the analytical approach proposed by Eckart[43] and further developed by Bell[44] (see details in Supplementary Methods and Supplementary Figs. 18–21). Such short transfer distances / high frequencies result into a tunneling effect reducing the reaction barriers by ~50 % for ThiH from 20.0 to 9.6 kcal.mol$^{-1}$ (Fig. 4a, Supplementary Fig. 19 and Supplementary Table 4) and by ~40 % for NosL from 18.3 to 10.7 kcal.mol$^{-1}$ (Fig. 4b, Supplementary Fig. 19 and Supplementary Table 4). Calculations were performed at 60 °C (*Tc*ThiH) and 25 °C (*Sa*NosL), corresponding to the optimal growth of *T. carboxidivorans* and *S. actuosus*, respectively.

**L-tyrosine Cα–Cβ bond break at the ThiH active site.** The step leading to *p*-cresyl• and DHG (Fig. 1) was next modeled from the [L-Tyr-NH•] intermediate discussed in the previous section (Fig. 4a). The reaction energy profile is shown in Fig. 5. Strikingly, despite the expected stability of the *p*-cresyl•[15] due to the electronic delocalization over the whole molecule, the [*p*-cresyl• + DHG]-*Tc*ThiH model was found to be only 1 kcal.mol$^{-1}$ below the transition state. In fact, it appears clear that the scission reaction is not complete at the QM/MM level of calculation since there is still a non-negligible part of the radical spin on the amino nitrogen (Fig. 5) and Cα and Cβ are only 2.65 Å apart. It also reflects greater compactness along the Cα–Cβ direction in ThiH (and in NosL) compared to the Cα–C direction in NosL, for

which we modeled a complete product formation[37]. If we try to separate them by scanning the Cα–Cβ bond further, the energy always increases, consistent with the cavity calculation on the X-ray model showing that L-tyrosine is tightly embedded in the active site cavity (Fig. 3c, left).

However, a product escape route is observed through a gate defined by residues Y76 and Y181 (Fig. 3c, left) and the thin channel that connects the active site to the surface is equivalent to the one observed in *Ti*HydG (Fig. 3c, center); the difference is that, in *Ti*HydG, Y76 and Y181 are replaced by two phenylalanine residues (F81 and F192 in Fig. 3c, center). In *Ti*HydG, the channel connects the active site to a second reaction site in the protein, where DHG is further processed into carbon monoxide and cyanide[35]. Early biochemical characterizations of ThiH from *E. coli* reported the existence of a complex between ThiH and ThiG to avoid DHG hydrolysis upon transfer[11]. It is therefore probable that, like in HydG, DHG would be transferred through the observed channel toward the bottom of the β-barrel. This channel, which is surrounded by polar residues and filled with water molecules, is absent in NosL (Fig. 3c, right). Given that QM/MM calculations are *de facto* performed at 0 K, normal thermally-activated movements cannot be observed. Hence, we decided to use the 250-ns MD simulation of L-tyrosine bound *Tc*ThiH we had performed at 333 K (see X-ray structure analysis above) to extract a sampling of active site conformations corresponding to normal structural "breathing" in order to determine whether some of them could separate the products further. As a reference, we first scanned the Cα–Cβ bond break using a reduced quantum model consisting of about 300 atoms extracted from our QM/MM model of [L-Tyr-NH•], and observed that the energy reaches a minimum when Cα and Cβ are about 2.7 Å apart but goes back up at greater scanned distances (red line with triangles curve in Fig. 6a). If we then geometry-optimize the last point of the scan where the Cα–Cβ distance is set at 3.1 Å, it goes back to 2.8 Å, but not lower, because of the flat potential in the region, and the spin analysis shows some residual spin on the amino nitrogen. This behavior is consistent with our QM/MM model (Fig. 5). In contrast, all equivalent reduced quantum model constructed from the MD frames we sampled show a lowering of the energy as the scanned Cα–Cβ distance increases and after geometry-optimizing the last point of these models, they all converged to a Cα–Cβ distance greater than 3.0 Å with the spin density located exclusively on the

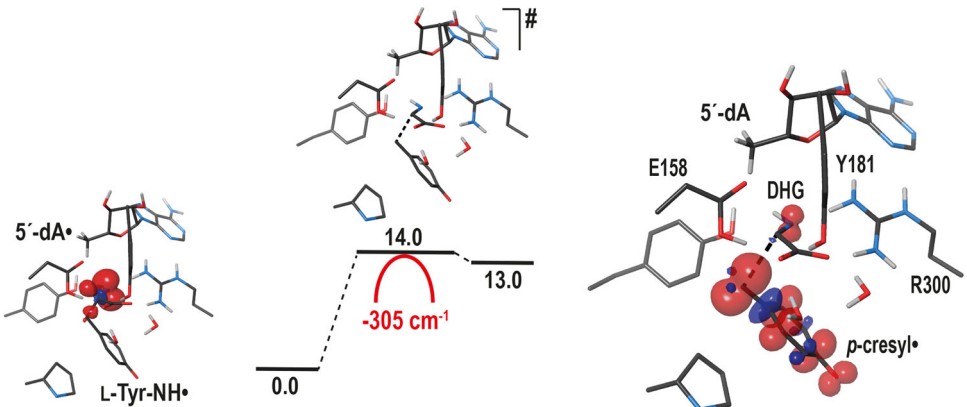

**Fig. 5 L-tyrosine Cα–Cβ bond break in *Tc*ThiH.** Reaction energy profile for the Cα–Cβ scission of L-Tyr-NH• leading to the formation of *p*-cresyl• and DHG. Single point energies for the reactant, transition state and product are given in kcal.mol$^{-1}$; the value of the TS single imaginary frequency is indicated in red; for the reactant, transition state and product, only a minimal model of the active site extracted from the corresponding QM/MM geometry is represented as sticks with C, N, O and H atoms in gray, blue, red and light gray respectively; only polar hydrogens are shown for clarity; the spin density is shown for both the reactant and product.

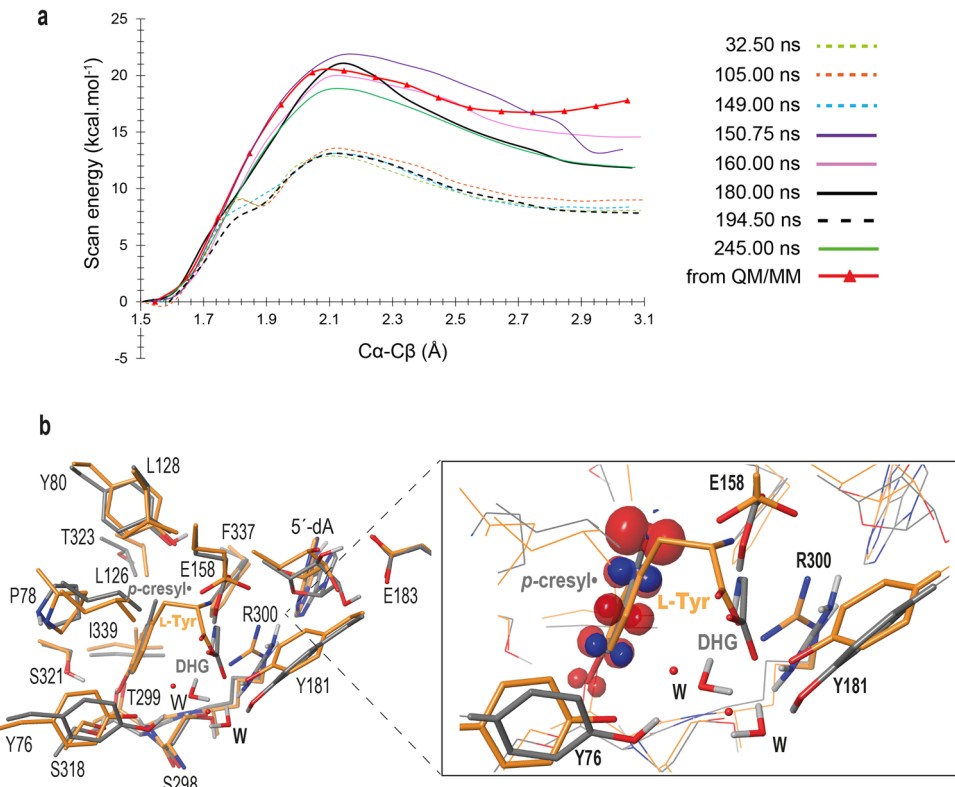

**Fig. 6 Release of L-tyrosine cleavage products. a** Cα–Cβ distance scan in quantum models restricted to about 300 atoms extracted along the trajectory of the 250-ns MD simulation of L-Tyr bound *Tc*ThiH (see Supplementary Methods); the corresponding simulation time is given for each curve from 32.5 to 245 ns. The scan depicted as a red line with red triangles was calculated on the same 300 atoms but extracted from the QM/MM product model we obtained (Fig. 4a). **b** The crystal structure of *Tc*ThiH active site (C, N, O, H in orange, blue, red and light gray, respectively) is superimposed to the structure corresponding to the geometry optimization of the last point of the scan of the 180-ns products (C, N, O, H in gray, blue, red and light gray, respectively). In the zoom of the L-Tyr (and products) binding site, the spin density is shown for the 180-ns product structure.

*p*-cresyl• moiety, hence supporting a completed cleavage reaction. The largest separation found for the 180-ns frame at 3.4 Å is shown in Fig. 6b. Remarkably, superimposing this model with the starting crystal structure indicates that they are very similar. The main movements reside at the location of residues Y76 and Y181 at the channel gate (Fig. 3c, left) and that, while the *p*-cresyl• moiety stays at the location of the L-tyrosine ring, the main movement is at the amino moiety upon DHG formation. However, we do not observe the substantial energy drop we had anticipated because of the expected *p*-cresyl• stability[15]. Furthermore, a 250-ns MD simulation at 333 K of the modeled *p*-cresol and DHG in the *Tc*ThiH active site did not result in the product escape, suggesting that other conformational changes (beyond the scope of this work) are necessary for product release.

## Discussion

The crystal structure of L-tyrosine-bound *Tc*ThiH has shown that the substrate acquires an unusual protonation state upon its binding at the active site. Indeed, the phenol functional group of L-tyrosine is deprotonated and the resulting phenolate stabilized by four anion-dipole interactions with strictly conserved residues in ThiH but also in the tyrosine lyase HydG (Figs. 2d, 3a). The active site structure of the substrate-free ThiH favors a lock-and-key model for L-tyrosine binding. This is so because no substantial changes are observed between the substrate-bound and free structures (Fig. 2c, f, respectively). In addition, as predicted when we solved the L-tryptophan bound NosL structure[25], this new structure confirms that 5′-dA• attacks the amino group of

L-tyrosine in ThiH (Fig. 3b). HydG shares this catalytic step in accordance with their active site homology. Furthermore, our modeling of the 5′-dA• hydrogen abstraction in ThiH and NosL reveals that the compactness of the site brings 5′-dA• very close to the amino group, inducing hydrogen tunneling for both enzymes; this can probably be generalized to all radical SAM enzymes given the similar compactness also observed in available [5′-dA]–substrate bound structures[45,46] and in [SAM]-substrate bound structures[13,47]. It also shows that, despite exhibiting conserved active sites, the sum of tiny structural differences is likely to be responsible for subtle changes during this step that will orient the reaction toward the subsequent Cα–Cβ or the Cα–C bond break in ThiH / HydG and NosL, respectively.

The reaction path for the Cα–Cβ bond break in [L-Tyr-NH•] in ThiH (Fig. 1) could not be completed using QM/MM calculations (Fig. 5), probably because additional conformational changes are required. Quantum calculations on active site models extracted from MD simulations confirmed that slight structural changes unlock the C–C bond cleavage and revealed that the *p*-cresyl• moiety remains in place at the position adopted by the L-tyrosine side-chain, while DHG has more space to move further away from *p*-cresyl• (Fig. 6). DHG being spontaneously hydrolyzed into glyoxylate in water[10], it must be protected from solvent upon its transfer from ThiH to ThiG. Early biochemical characterization of ThiH from *E. coli* reported the existence of a complex between ThiH and ThiG[11]. Hence, one cannot exclude structural changes induced by ThiG are necessary to trigger DHG production and transfer. Conversely, the absence of ThiG might prevent the transfer to avoid DHG loss. Our observation that the C-terminal stretch can adopt at least two different conformations,

covering or not the β-barrel, supports the idea that DHG transfer is occurring, in both ThiH and HydG, through the observed channel defined by residues Y76/Y181 and F81/F192, respectively (Fig. 3c, left and center, respectively). In addition, the specific interaction between the carboxyl moieties of L-tyrosine and residue E158 (or E133 in HydG) might lock further motions. Proton transfer from the protonated glutamic acid residue to DHG might also participate in its release and safe transfer to ThiG. Taken together our results show that hydrogen tunneling lowers the energy barrier to afford efficient substrate activation by radical SAM enzymes. The comparison between ThiH and NosL unveils how slight structural differences optimized over the evolution process led to changes in substrate and reaction selectivities in enzymes otherwise very close. These findings will certainly pave the way to a better understanding of the structure-function relationships taking place in radical SAM enzymes and will contribute to the engineering of these enzymes for future biotechnological applications.

## Methods

**Expression and protein purification**. The synthetic gene of ThiH from *Thermosinus carboxydivorans* was purchased from GenScript™. The DNA sequence was codon optimized for expression in *Escherichia coli* and inserted into a pET-15b plasmid (Novagen™) between the NcoI and BamHI restriction sites, leading to a protein containing an N-ter Strep-tag and corresponding to the pTcThiH construct. The corresponding DNA sequence is as follow:

CC**ATG**GGACATCATCATCATCATCACTCCGGCACGTTCTATGATGTG
ATTGAAGATTACCGCCACTTTGATTTCGCAGCGTATTTCGCGAAAGTTA
CCGATTCTGACGTCCGTCGCATCCTGCGTCAGGATCGCCTGTCAGCC
CTGGACTTTCTGACGCTGCTGTCGCCGCAAGCGGAAGCCTATCTGGAAG
AAATGGCACAGAAAGCTCATCGTCTGACCGTTCAACACTTTGGCCGCA
CCATGCTGCTGTATACGCCGCTGTACCTGGCGAACTATTGCGTGAATCA
GTGCGTTTACTGTGGTTTCCAACTGAAAAACAAACTGGAACGTAAAA
AACTGACCCTGGCGGAAGTGGAACAGGAAGCCCAACTGATTGCGGCCAC
CGGCCTGAAACATATTCTGATCCTGACGGGTGAAAGTCGCCAGCACTC
ACCGGTCTCGTATATCAAAGATTGTGTGAACATCCTGAAAAAATACTTT
AGCTCTATCAGCATCGAAATTTATCCGCTGACCCAGGAAGAATACGCAG
AACTGATCGGCGCTGGTGTTGACGGCCTGACGATTTACCAGGAAGTGT
ATAACGAAGAAGTTTATGCGGAAATGCATCCGGCCGGCCCGAAACGT
AATTACCGTTTCCGCCTGGAAGCACCGGAACGTGCATGCCAGGCCGGT
ATGCGTACCGTGAACATCGGCGCTCTGCTGGGTCTGAATGATTGGCGCC
AGGAAGCATTTTTCACGGGTCTGCATGCTGATTATCTGCAACGTCGC
TTTCCGGACGTGGAAGTTAGTATTTCCCCGCCGCGTATGCGTCCGCACCT
GGGCGGTTTCCCGCCGCGTGTGGTTGTCAGCGATCAGAACCTGGTCCA
ATATGTGCTGGCATTTCGTCTGTTCATGCCGCGTAGTGGTATCACCCTGT
CCACGCGTGAAAATGGTCGTCTGCGTGACGCGATGGTCCGTCTGGGCG
TGACCAAAATGAGCGCAGGTTCTTGTACGGCTGTTGGCGGTCGTAGCG
ATCAGGAAGCCGTCGGCCAGTTCCAAATTTCTGACGAACGCACCGTTGC
GGAAGTCGCAGCTATGCTGTATGCCGCCCAGGGTTATCAACCGGTGTACAAA
GATTGGCAGGCAC**TGA**GGATCC

which leads to the following protein sequence:

MWSHPQFEKASGTFYDVIEDYRHFDFAAYFAKVTDSDVRRILRQDRLSA
LDFLTLLSPQAEAYLEEMAQKAHRLTVQHFGRTMLLYTPLYLANYCVNQCVY
CGFQLKNKLERKKLTLAEVEQEAQLIAATGLKHILILTGESRQHSPVSYIKDC
VNILKKYFSSISIEIYPLTQEEYAELIGAGVDGLTIYQEVYNEEVYAEMHPAG
PKRNYRFRLEAPERACQAGMRTVNIGALLGLNDWRQEAFFTGLHADYLQ
RRFPDVEVSISPPRMRPHLGGFPPRVVVSDQNLVQYVLAFRLFMPRSGITLST
RENGRLRDAMVRLGVTKMSAGSCTAVGGRSDQEAVGQFQISDERTVAEVA
AMLYAQGYQPVYKDWQAL.

The recombinant *thiH* gene was coexpressed with the *isc* operon and the SAM synthase gene coding *metK* from *E. coli* using an *E. coli* BL21(DE3) strain containing both pTcThiH and our home-made pRSF-ISC-MetK plasmids[25]. Cells were grown aerobically at 37 °C under agitation in 4 L of TB medium supplemented with 50 mM MOPS/KOH, pH 7.1, 0.5% D-glucose, ampicillin (100 μg.ml⁻¹) and kanamycine (50 μg.ml⁻¹) up to an $OD_{600}$ of about 0.6. They were subsequently transferred in 2 L sealed bottles into a glove box with an anaerobic atmosphere containing less than 5 ppm $O_2$ under constant stirring at 20 °C. After about 15 min, the bottles were opened to equilibrate the medium with the atmosphere of the glove box and the medium was supplemented with 10 mL MEM Vitamin solution 100X (SIGMA M6895); 4 g fumarate; 500 μL 0.5 M cysteine and 750 μL 0.2 M L-methionine per L of culture. Fifteen more minutes later, the medium was enriched with 250 μL 1 M ammonium iron(III) citrate. Protein expression was induced when the $OD_{600}$ reached 1 with addition of 1 mM final concentration isopropyl-β-D-1-thiogalactopyranoside and cells were further grown overnight before being harvested and frozen in liquid nitrogen. All the purification steps were performed under anaerobic conditions. Cell pellets were resuspended in

the glove box in buffer A (50 mM Tris pH 8; 150 mM NaCl) supplemented with EDTA-free protease inhibitor cocktail. They were subsequently disrupted by sonication and the crude-extract was cleared by centrifugation in sealed tubes at 15,000 × rpm during 30 min at 4 °C. The clarified supernatant was filtered, and loaded onto a streptavidin-agarose column (Iba®) equilibrated with anaerobic cleared lysate was injected into a Strep-tactin column equilibrated with buffer A. After an extensive wash step, the protein was eluted with a 1 mM desthiobiotin solution in buffer A. The iron-sulfur cluster was subsequently reconstituted. The protein was incubated 15 min with 5 mM DTT prior to the addition of 200 nM NifS from *Azotobacter vinelandii*. 5 excess $FeCl_3$ and L-cysteine were then added and the solution was kept under mild stirring overnight at 20 °C. The protein was further purified to homogeneity by gel filtration using a GE-healthcare highload 16/600 S200 prep-grade column equilibrated with buffer B (50 mM Tris pH 8 and 100 mM NaCl). The protein eluted as two peaks corresponding to a dimer and a monomer, respectively. Only the fractions corresponding to the monomer (major peak) were pooled, concentrated to 10 mg/mL and stored in liquid nitrogen. Iron content was determined using the method of Fish[48] and indicated over 3.9 iron equivalents per ThiH molecule.

**Crystallization and X-ray structure determination**. Crystals were obtained using the vapor diffusion method. Initial conditions were determined by screening 1248 conditions using a Gryphon robot (Ari Instruments, CA, USA) setup in a glove box and were subsequently manually optimized. Crystals suitable for X-ray diffraction experiments were obtained with 1.4–1.6 M $(NH_4)_2SO_4$, 100 mM MES buffer pH 6.5, 15% dioxane and 10 mg/mL *Tc*ThiH. The protein was preincubated with 1 mM 5′-dA and 1 mM L-methionine. When required, 5-excess L-tyrosine was added to the crystallization drop. Crystals were cryoprotected using 25% glycerol in addition to the crystallization condition and were subsequently mounted in cryoloops before flash cooling using liquid propane in the glove box[49]. Data were collected at SOLEIL at beamlines PROXIMA-1 and PROXIMA-2A and were processed using the XDS package[50] (see Supplementary Table 1). Crystals belong to the space group C2 with two molecules per asymmetric unit. The crystal structure was solved by molecular replacement using the program PHASER[51] and the crystal structure of NosL from *Streptomyces actuosus* (PDB ID 4R34) as a search model. The initial model was manually corrected using COOT[52] and refined using PHENIX[53] to an $R_{work} = 0.153$ and $R_{free} = 0.174$ with good geometry and over 98% residues in the most allowed region of the Ramachandran plot[54]. This model was subsequently used to solve the ligand-free *Tc*ThiH crystal structure following the same procedure, leading to a model with good geometry. The corresponding statistics are reported in Supplementary Table 1.

**Theoretical calculations**. Details of all calculations are given in the Supplementary Information. All MD simulations, QM and QM/MM calculations were performed within the Schrödinger suite[55].

**Reporting summary**. Further information on research design is available in the Nature Research Reporting Summary linked to this article.

## Data availability

Structure coordinates and diffraction data of the L-tyrosine-bound and the substrate-free *Tc*ThiH were deposited at the Protein Data Bank (http://www.pdb.org) under accession codes 7PD1 and 7PD2, respectively. Source data are provided with this paper. The data that support the findings of this study are available from the corresponding author upon request.

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

## Acknowledgements

We thank Dr Pierre Legrand from SOLEIL for his help with data collection and our colleague Dr Anne Volbeda for an updated version of the CAVENV cavity calculation program he developed. This work was supported by the Radis-Bio contract from the CEA/DRF-Impulsion program and the CARBONARA contract (YN) from the Agence Nationale pour la Recherche (ANR-16-CE29-0019). Part of this work used the platforms of the Grenoble Instruct-ERIC center (ISBG; UMS 3518 CNRS-CEA-UGA-EMBL) within the Grenoble Partnership for Structural Biology (PSB), which is supported by FRISBI (ANR-10-INBS-05-02) and GRAL and financed within the Université Grenoble Alpes graduate school (Ecoles Universitaires de Recherche) CBH-EUR-GS (ANR-17-EURE-0003). We appreciate the help from the staff of the computing facility provided by the Commissariat à l'Energie Atomique et aux Energies Alternatives (CEA/DRF/GIPSI), Saclay, and CCRT, Bruyères-le-Châtel.

## Author contributions

Y.N. conceived and directed the project. C.S. and L.M. expressed, purified and crystallized the protein *Tc*ThiH. Y.N. collected the diffraction data, solved and analyzed the structure. P.A. and J.-M.M conceived and performed the calculations. P.A., J.-M.M. and Y.N. wrote the manuscript.

## Competing interests

The authors declare no competing interests.
