## [Peer Review File · Nature Communications]

REVIEWER COMMENTS

Reviewer #1 (Remarks to the Author):

The manuscript "Crystal structure of L-tyrosine-bound ThiH reveals the origin of C-C bond break differences within radical SAM aromatic amino acid lyases" by Nicolet and coworkers reports the X-ray structure of tyrosine bound ThiH. The authors perform structural comparisons and QM/MM computational analyses amongst ThiH, HydG (a tyrosine lyase), and NosL (a tryptophan lyase). These data demonstrate that substrate binding to ThiH (and presumably HydG) is accomplished via the fully deprotonated form (amine, carboxylate, and phenol groups) of L-tyrosine. This substrate binding mode in conjunction with active site compactness appears to promote a hydrogen tunneling mechanism from the substrate amine group to the deoxyadenosyl radical intermediate that is formed following cleavage of S-adenosylmethionine.

The work carries additional significance because it provides insight into the reaction paths that lead to C α -C β bond cleavage in ThiH/HydG (tyrosine) and C α -carboxylate bond cleavage in NosL (tryptophan). While QM/MM is outside the expertise of this reviewer, I find the presentation of this data to be fairly clear and compelling. While no sizeable distinctions explicitly demonstrate reasons for the two reaction paths, quantum analysis of the MD frames/snapshots support the idea that subtle perturbations within the active site may concertedly impact the chemistry and dissolution of dehydroglycine formation. The complexity is underscored by the presence of internal product channels within the TIM barrel core wherein DHG migrates, presumably to either be transferred to ThiG for further processing or degraded into cyanide and carbon monoxide by HydG; NosL lacks an internal cavity and thus appears simpler to model. The details of either DHG or CO/CN transfer to downstream proteins are not well defined in the literature, but the data reported herein speak to the idea that protein-protein associations could influence the active site structure in such a manner as to facilitate product release into the internal cavity. I suppose this would be an intriguing aspect for future studies. The manuscript is well written and the experiments seem adequate to address the fundamental mechanistic questions raised by these three aromatic amino acid lyases. The figures are clear and descriptive, although it would be beneficial if Figure 2 panels b and c were larger to more easily see the fine detail present here. For these reasons, I support publication of the work and have only minor suggestions.

Comments:

1. The conservation of the internal channels in ThiH and HydG is remarkable, even to the two aromatic amino acids that gate entry (Y76 and Y181 in ThiH and F81 and F192 in HydG). You mention in Fig. 2 legend that the cavity maps were contoured with a probe radius of 1 angstrom. In the discussion (lines 459 – 462) you say that the internal cavity is for DHG transfer/migration. I wonder if it would be appropriate to add a sentence here describing the nature of this cavity. I presume it is hydrophobic in nature, and clearly it is small given the use of a 1 angstrom probe radius. Are the characteristics of the two internal channels in ThiH and HydG similar? Perhaps this has been discussed elsewhere in the author's work, but some mention of this would contribute to the discussion in this section.

2. Along these lines, would it be possible to simulate how dynamic this channel is in relation to DHG migration via MD calculations? I don't know the difficulties of an in silico experiment like this, and certainly appreciate that such an idea is outside the scope of the current body of work. This said, any information regarding how DHG is mobilized from the radical SAM cluster active site, and into and down the internal channel to the bottom of the TIM barrel would be highly beneficial for researchers in the field. It would help us to understand or appreciate how ThiH protects DHG against hydrolysis, as well as how HydG degrades this molecule into cyanide and carbon monoxide. This is especially important for HydG in terms of existing models in the literature for synthon formation. For example, how would synthon formation and release impact the structure of HydG and its internal cavity? This is something that has not been addressed in

the literature. Again, I understand that this is outside the scope of the current work but since the author's are experts in this field and have published on HydG, I am interested if any thought has been given to this.

3. In the discussion section, line 431 the authors state "...phenol function of L-tyrosine is deprotonated...". The word "function" here is awkward, consider replacing it with "group" or "functional group".

4. Line 454. The authors talk about the spontaneous hydrolysis of DHG into glyoxylate and water. Please provide a reference for this.

5. Line 457. The authors state: "...structural changes induced by ThiG and necessary to trigger DHG production....", replace "and" with "are".

6. Supplemental figures, general comment. The legends for several of these simply refer to main body figures. While this is fine, it would generally assist readers if you could provide a brief description to what the particular SI figure is showing. This would be helpful since the manuscript describes three different enzymes (ThiH, HydG, NosL) in substrate bound and substrate free states.

7. Supplementary Figure 8, legend. For ease, please denote the channel colors' relation to the enzyme source as was done in Figure 2c. This would be helpful since the order of the images in this figure are different than what is presented in Figure 2c (in other words, this will save the reader time in referring back to the main body).

Reviewer #2 (Remarks to the Author):

This is a very interesting manuscript where the authors have performed a nice job. The possible criticisms are related to the calculations performed, which only consider potential energies: only potential energy surfaces have been explored, and no statistical treatments have been carried out to estimate free energies. The consequence of this is that one is limited by the reaction path one has located, which may not be the most representative for describing the chemical process studied... (an example of this could be seen in Figure 5, where two sets of results seem to be obtained?).

On the other hand, the level of calculation used in the study makes practically unfeasible any other approach to the problem nowadays, apart from the one used by the authors. Finally, since the tunnel effect plays such an important role, it could have been evaluated using less approximate methods (such as small curvature tunnel or instanton theory), or contrasted against any of them. In any case, the range of kappas obtained ($10^5 - 10^6$) is within the range observed in some enzymatic systems.

Doubts:

line 4: "C bond" should not be "COO- bond"? (line 56).

line 394: is the sum of the partial charges of those 300 selected atoms close to the value seen by the QM atoms in the full calculations?

Supplementary material: page 5

How were the windows chosen from which the geometries were extracted to perform the scans? (ie, why these values of 32.5, 105, ..., 245 ns?)

Reviewer #3 (Remarks to the Author):

The manuscript by Amara et. al., entitled "Crystal structure of L-tyrosine-bound ThiH reveals the origin of C-C bond break differences within radical SAM aromatic amino acid lyases" details the structure of ThiH. The authors crystallize this protein and solve its structure with L-tyrosine bound. They then compare this structure to a related radical SAM protein NosL. Using computational studies, the authors attempt to pinpoint the structural differences that account for the reaction specificity of these enzymes. The structural work presented in this manuscript is well-done. In my opinion, the manuscript suffers a bit from a lack of figures, the small size of figures that are presented, and a lack of descriptive SI legends.

Overall, the paper is intriguing in that it points out the small structural differences in a subset of radical SAM proteins that catalyze different reactions. However, it seems the novelty of the work and the insights gained, do not provide a large advance in the mechanistic understanding of these enzymes.

Specific comments:

- 1. The SI figure legends should describe what is in figure and not simply refer to a main text panel. This document should stand on its own for readability.**
- 2. The statement in the abstract that says "we report the crystal structure of L-tyrosine bound ThiH from Thermosinus carboxydivorans revealing an unusual protonation state" needs to be rewritten. As described by Fisher (2011, Acta Crystallographic Section D), the protonation state of a residue using X-ray crystallography can be determined only in cases of ultra-high resolution. This statement should also be changed when it appears again in lines 68-69. The bond distances between the presumed deprotonated Tyr and the interacting residues need to be included in a figure and the coordinate error at the resolution determined should also be mentioned.**
- 3. The authors should experimentally show that there is a pH dependence on the rate of catalysis in this enzyme.**
- 4. Line 25: delete the**
- 5. Scheme 1: add space between the top set of reactions and the bottom because they are running together.**
- 6. Lines 45-54: HydG and ThiH share 27-percent sequence identity with each other and each share 25-percent sequence identity with NosL? Are these statements interpreted correctly? Line 222 says that ThiH and NosL share less than 23% identity.**
- 7. Add the reaction of HydG, ThiH, and NosL into the supplement.**
- 8. Add a supplementary figure that shows that comparison being made between BioB, HydE, PylB, HydG, and NosL. The comparison with NosL can be used to back up the statement that the active site of ThiH is reminiscent of NosL.**
- 9. Add a figure to show the described different conformations of chains A and B that includes the crystal contacts.**
- 10. In line with the above comment since the region is mentioned to be structurally flexible, include a discussion about the B-factors in this region.**
- 11. For Figure 1d, add a complementary panel that shows the maps of L-Tyr and the interacting residues.**
- 12. For Figure 1d, add the distances of the highlighted interactions.**
- 13. For the description of L-Tyr binding in the active site add references to Figure 1d if that is the figure that best shows the described interactions. See lines 136-137 and 137-138.**
- 14. The structures and labels in Figure 2 are too small.**
- 15. Add a figure that supports this statement: "HydG crystal structures display significant structural differences with ThiH".**
- 16. With the gained computational insight about the structure-function relationships in the reaction specificity in NosL and ThiH, it would be nice to include an experiment that shows that the reaction outcome can be changed (is there a mutagenesis reaction that could support the**

claims made?).

17. Lines 441-442 state that "this can probably be generalized to radical SAM enzymes given the similar compactness also observed in available 5'dA – substrate bound structures" references just AprD4 and HydE. Is this statement means to be generalized to the entire superfamily of radical SAM enzymes or just this small subset? The former is likely a stretch to base on the three mentioned structures so the statement should be made more clearly.

18. The abstract mentions that the two enzymes NosL and ThiH share significant primary structure similarities, but the sequence identity seems to be less than 30-percent which does not seem as significant as implied.

19. Line 80: "these enzymes work as assembly lines". I am unclear on the meaning of this sentence.

ANSWER TO THE REFEREES

Reviewer #1 (Remarks to the Author):

Reviewer: The manuscript “Crystal structure of L-tyrosine-bound ThiH reveals the origin of C-C bond break differences within radical SAM aromatic amino acid lyases” by Nicolet and coworkers reports the X-ray structure of tyrosine bound ThiH. The authors perform structural comparisons and QM/MM computational analyses amongst ThiH, HydG (a tyrosine lyase), and NosL (a tryptophan lyase). These data demonstrate that substrate binding to ThiH (and presumably HydG) is accomplished via the fully deprotonated form (amine, carboxylate, and phenol groups) of L-tyrosine. This substrate binding mode in conjunction with active site compactness appears to promote a hydrogen tunneling mechanism from the substrate amine group to the deoxyadenosyl radical intermediate that is formed following cleavage of S-adenosylmethionine.

The work carries additional significance because it provides insight into the reaction paths that lead to C α -C β bond cleavage in ThiH/HydG (tyrosine) and C α -carboxylate bond cleavage in NosL (tryptophan). While QM/MM is outside the expertise of this reviewer, I find the presentation of this data to be fairly clear and compelling. While no sizeable distinctions explicitly demonstrate reasons for the two reaction paths, quantum analysis of the MD frames/snapshots support the idea that subtle perturbations within the active site may concertedly impact the chemistry and dissolution of dehydroglycine formation. The complexity is underscored by the presence of internal product channels within the TIM barrel core wherein DHG migrates, presumably to either be transferred to ThiG for further processing or degraded into cyanide and carbon monoxide by HydG; NosL lacks an internal cavity and thus appears simpler to model. The details of either DHG or CO/CN transfer to downstream proteins are not well defined in the literature, but the data reported herein speak to the idea that protein-protein associations could influence the active site structure in such a manner as to facilitate product release into the internal cavity. I suppose this would be an intriguing aspect for future studies. The manuscript is well written and the experiments seem adequate to address the fundamental mechanistic questions raised by these three aromatic amino acid lyases. The figures are clear and descriptive, although it would be beneficial if Figure 2 panels b and c were larger to more easily see the fine detail present here. For these reasons, I support publication of the work and have only minor suggestions.

Answer: *We thank the Reviewer for his/her comments; Panels b and c of Figure 2 have now been enlarged for clarity.*

Comments:

Reviewer: 1. The conservation of the internal channels in ThiH and HydG is remarkable, even to the two aromatic amino acids that gate entry (Y76 and Y181 in ThiH and F81 and F192 in HydG). You mention in Fig. 2 legend that the cavity maps were contoured with a probe radius of 1 angstrom. In the discussion (lines 459 – 462) you say that the internal cavity is for DHG transfer/migration. I wonder if it would be appropriate to add a sentence here describing the nature of this cavity. I presume it is hydrophobic in nature, and clearly it is small given the use of a 1 angstrom probe radius. Are the characteristics of the two internal channels in ThiH and HydG similar? Perhaps this has been discussed elsewhere in the author’s work, but some mention of this would contribute to the discussion in this section.

Answer: *The channels are gated by aromatic residues in both ThiH and HydG. Yet the remaining residues are polar with no specific conservation. Supplementary Figure 11 displays stereoviews of the cavities for the reader to better assess their respective width. We have now added a sentence to better describe the polar properties of the surrounding residues and the presence of water molecules.*

Reviewer: 2. Along these lines, would it be possible to simulate how dynamic this channel is in relation to DHG migration via MD calculations? I don't know the difficulties of an in silico experiment like this, and certainly appreciate that such an idea is outside the scope of the current body of work. This said, any information regarding how DHG is mobilized from the radical SAM cluster active site, and into and down the internal channel to the bottom of the TIM barrel would be highly beneficial for researchers in the field. It would help us to understand or appreciate how ThiH protects DHG against hydrolysis, as well as how HydG degrades this molecule into cyanide and carbon monoxide. This is especially important for HydG in terms of existing models in the literature for synthon formation. For example, how would synthon formation and release impact the structure of HydG and its internal cavity? This is something that has not been addressed in the literature. Again, I understand that this is outside the scope of the current work but since the author's are experts in this field and have published on HydG, I am interested if any thought has been given to this.

Answer: *As we mentioned in the paragraph just before the Discussion section, a 250-ns dynamics of [*p*-cresol + dehydroglycine] did not result in product escape. We agree with the Reviewer that even if it is out of the scope of our study, it is a very important issue. Concerning HydG it is clear that understanding the mechanism of cyanide and carbon monoxide formation as well as the conformational changes that may or need to occur upon product release remain to be elucidated as we pointed out in a review recently published in ACS Bio&MedChem^{Au} (<https://doi.org/10.1021/acsbioimedchemau.1c00044>) that is now cited in the main text.*

*During the review process of our manuscript, a quantum chemical study on the HydG mechanism was published in Biochemistry (<https://doi-org.insb.bib.cnrs.fr/10.1021/acs.biochem.1c00379>) suggesting a radical relay mechanism. However, this publication raises many questions and discussing them in the present manuscript is out of the scope of our study and would blur the present message. Indeed, they do not address the radical transfer along the channel and propose a direct radical relay between *p*-cresyl• and DHG. Yet, in a second publication, the same authors report on the accumulation of an oxidobenzyl radical (*p*-cresyl•), which is in contradiction with their proposed radical transfer from *p*-cresyl• to DHG without discussing further this discrepancy between their calculations and their experiments. Obviously further experimental investigation is needed to move forward on these questions.*

Reviewer: 3. In the discussion section, line 431 the authors state "...phenol function of L-tyrosine is deprotonated...". The word "function" here is awkward, consider replacing it with "group" or "functional group".

Answer: *Done.*

Reviewer: 4. Line 454. The authors talk about the spontaneous hydrolysis of DHG into glyoxylate and water. Please provide a reference for this.

Answer: *Done.*

Reviewer: 5. Line 457. The authors state: "...structural changes induced by ThiG and necessary to trigger DHG production....", replace "and" with "are".

Answer: *Done.*

Reviewer: 6. Supplemental figures, general comment. The legends for several of these simply refer to main body figures. While this is fine, it would generally assist readers if you could provide a brief description to what the particular SI figure is showing. This would be helpful since the manuscript describes three different enzymes (ThiH, HydG, NosL) in substrate bound and substrate free states.

Answer: Done. All SI legends were checked and completed as suggested by Reviewers 1 and 3.

Reviewer: 7. Supplementary Figure 8, legend. For ease, please denote the channel colors' relation to the enzyme source as was done in Figure 2c. This would be helpful since the order of the images in this figure are different than what is presented in Figure 2c (in other words, this will save the reader time in referring back to the main body).

Answer: Supplementary Fig. 8 (now Supplementary Fig. 11) has been modified to match the image order of corresponding Fig. 2c and a more explicit legend has been added to also address Reviewer 3's comment.

Reviewer #2 (Remarks to the Author):

Reviewer: This is a very interesting manuscript where the authors have performed a nice job. The possible criticisms are related to the calculations performed, which only consider potential energies: only potential energy surfaces have been explored, and no statistical treatments have been carried out to estimate free energies. The consequence of this is that one is limited by the reaction path one has located, which may not be the most representative for describing the chemical process studied... (an example of this could be seen in Figure 5, where two sets of results seem to be obtained?). On the other hand, the level of calculation used in the study makes practically unfeasible any other approach to the problem nowadays, apart from the one used by the authors.

Answer: We agree with the Reviewer, a statistical treatment appeared extremely difficult to us given the complexity of the system. The calculations described in Figure 5 were performed to try to circumvent this limitation and appreciate if protein dynamics could help separate both products i.e. p-cresyl• and dehydroglycine. Given the computational time required for the QM calculations that followed the MD simulation, we extracted only a few frames. The fact that we found at least two sets of results is an indication that dynamics indeed will allow product separation which was the purpose of this calculation.

Reviewer: Finally, since the tunnel effect plays such an important role, it could have been evaluated using less approximate methods (such as small curvature tunnel or instanton theory), or contrasted against any of them. In any case, the range of kappas obtained (10^5 - 10^6) is within the range observed in some enzymatic systems.

Answer: The « small-curvature tunneling approximation » (SCT) (<https://doi.org/10.1021/j150621a001>) is a multidimensional reaction path curvature theory that aims at calculating the amount of « corner cutting » shortening the tunneling path. To operate it requires however the calculation of energy, gradient, and Hessian information along the reaction path, an information which is computationally costly and beyond the scope of our paper. We have now added a sentence and references in the Supplementary Methods. Indeed, our goal was only to show that, semi-quantitatively, the asymmetric Eckart model modeling the minimal energy path defined from R to P via TS already yields a significant lowering of the barrier (a rough factor 2 for both ThiH and productive reactant NosL (cf. Supplementary Table 4). Moreover, already at this level of analysis, the range of kappa values we obtained ($9 \cdot 10^4 \text{ s}^{-1}$ for productive reactant NosL and $6 \cdot 10^5 \text{ s}^{-1}$ for ThiH; cf. Eqs. S13-

S14) are well within the proper range of what is observed for some first-order ($R \rightarrow P$) enzymatic systems (see Table 1 in <https://doi.org/10.1126/science.7809611>), as notified by the reviewer. To illustrate this last point, we added a column to Supplementary Table 4 (k values in s^{-1}).

Alternatively, the Instanton theory (<https://doi.org/10.1063/1.430676>, <https://doi.org/10.1103/PhysRevD.16.1762>) aims at evaluating rate constants including tunneling effect by searching for an optimal tunneling path instead of a minimal energy path characterized by R , TS and P . But the drawback is the same as before for SCT: it also relies on the determination of a large number of energy and gradient calculations. Let us notice here that the instanton mechanism typically occurs below a crossover temperature (defined for example in M J Gillan, *J. Phys. C: Solid State Phys.* 20, 3621 (1987)) given by $T_c = h\nu/(2\pi k_B)$. This is the temperature below which tunneling becomes the dominant transition mechanism. In our case, with $\nu \sim 2200 \text{ cm}^{-1}$, we find $T_c \sim 500 \text{ K}$. The instanton path is thus expected to deviate from the classical reaction path (the IRC) at temperatures lower than T_c . We can therefore state that the effective barrier values we determined in Supplementary Table 4 using the Eckart approach are most probably upper limits. Thus, tunneling is operative at our level of calculation and would be even more efficient including the instanton mechanism.

Reviewer: Doubts:

Reviewer: line 4: "C bond" should not be "COO- bond"? (line 56).

Answer: *C* the standard atom name of the carbon in the L-tryptophan COO- group, as opposed to the L-tyrosine $C\beta$ atom we are referring to, in the same sentence, in the reaction catalyzed by ThiH. Thus, it has been corrected to *C* on line 56 and in two other locations in the main text as we are referring to one bond that is broken. The new Supplementary Scheme 1 requested by Reviewer 3 depicts the reactions catalyzed by ThiH, HydG and NosL, notably this difference in the bond that is cut after the [L-Tyr-NH•] and the [L-Trp-NH•] radicals are formed.

Reviewer: line 394: is the sum of the partial charges of those 300 selected atoms close to the value seen by the QM atoms in the full calculations?

Answer: *The reviewer raises the important point of the electrostatic effect and the potential inaccuracies of its modeling in our comparison of our QM/MM-derived and QM-derived results that could come from different total charges. In our QM/MM calculations, the QM part that consists of about 100 atoms with a total charge of -2, the rest of the system is treated with classical charges. In our QM calculations, we took these 100 atoms and added 200 atoms of the first and second coordination spheres. The resulting system has also a total charge of -2; thus the sum of the partial charges of the 'extra' 200 atoms treated with QM have a total neutral charge. Thus, neutrality and the large number of atoms we took, minimize the effect of the absence of the whole protein matrix in the QM calculation.*

Reviewer: Supplementary material: page 5 How were the windows chosen from which the geometries were extracted to perform the scans? (ie, why these values of 32.5, 105, ..., 245 ns?)

Answer: *The Reviewer is right, the choice of the windows was not described in the Supplementary Methods it has now been added. Since the problem we encountered in our calculations was the separation of the products, i.e. dehydroglycine and p-cresyl•, we looked for conformations along the dynamics where tyrosine residues 76 and 181 (please see the zoom of Figure 5b) have the largest displacement compared to the crystal structure. Indeed, dehydroglycine being held by arginine 300 and glutamate 158, we looked for conformations where room was made for the p-cresyl• to get further away from dehydroglycine. Ideally, a statistics on all MD frames would have been performed but as mentioned by the Reviewer the complexity of the system made such a calculation unfeasible.*

Reviewer #3 (Remarks to the Author):

Reviewer: The manuscript by Amara et. al., entitled “Crystal structure of L-tyrosine-bound ThiH reveals the origin of C-C bond break differences within radical SAM aromatic amino acid lyases” details the structure of ThiH. The authors crystallize this protein and solve its structure with L-tyrosine bound. They then compare this structure to a related radical SAM protein NosL. Using computational studies, the authors attempt to pinpoint the structural differences that account for the reaction specificity of these enzymes. The structural work presented in this manuscript is well-done. In my opinion, the manuscript suffers a bit from a lack of figures, the small size of figures that are presented, and a lack of descriptive SI legends.

Overall, the paper is intriguing in that it points out the small structural differences in a subset of radical SAM proteins that catalyze different reactions. However, it seems the novelty of the work and the insights gained, do not provide a large advance in the mechanistic understanding of these enzymes.

Specific comments:

1. The SI figure legends should describe what is in figure and not simply refer to a main text panel. This document should stand on its own for readability.

Answer: *We have completed all the SI figure legends as suggested by the Reviewers 1 and 3. In addition, we have added/modified all the figures requested by the Reviewer, hoping that the new version of the manuscript is now clearer.*

Reviewer: 2. The statement in the abstract that says “we report the crystal structure of L-tyrosine bound ThiH from *Thermosinus carboxydivorans* revealing an unusual protonation state” needs to be rewritten. As described by Fisher (2011, Acta Crystallographic Section D), the protonation state of a residue using X-ray crystallography can be determined only in cases of ultra-high resolution. This statement should also be changed when it appears again in lines 68-69. The bond distances between the presumed deprotonated Tyr and the interacting residues need to be included in a figure and the coordinate error at the resolution determined should also be mentioned.

Answer: *As crystallographers, we are of course aware of the fact that hydrogens can only be determined at ultra-high resolution but a protonation state can be deduced from the interactions observed in a crystallographic model, notably when high-resolution structures provide confidence in these interactions. Therefore, we consider the word “revealing” is the most appropriate one. Yet, lines 68-69 have been rewritten to clarify this point. The unusual geometry of the active site highlights a direct interaction between the N-amino nitrogen of L-tyrosine and the guanidinium of Arg300, the carboxyl moiety of L-tyrosine and the carboxyl of Glu158 and a phenol(ate) interacting with 4 hydrogen bond donors at suitable distances. Hence, either the amino group or the guanidinium must be deprotonated, which is unusual; either the carboxyl or Glu158 must be protonated, this is also unusual and the phenol might be deprotonated. These deductions from the crystallographic model were confirmed by the QM/MM calculations. In this particular case, all protein heavy atoms interacting with the O atom of the L-tyrosine hydroxyl group clearly indicate that it should be deprotonated. This was confirmed by our calculations including hydrogens in our models and testing both protonation states; as expected, only the deprotonated state reproduces the X-ray model. Furthermore, deprotonation of the phenol is in full agreement with a very recent publication by RD Britt and co-workers showing that the ensuing *p*-cresyl• intermediate is also deprotonated (4-oxidobenzyl radical - <https://doi.org/10.1021/acs.biochem.1c00619>). The bond distances have been indicated in figure 1d*

as requested by the reviewer. The estimated coordinate errors are now included in the Figure 1 caption as well. The crystal structure has been determined at 1.27 Å resolution. L-tyrosine and the active site are embedded. Therefore, the estimated errors are about 0.036 Å for all the L-tyrosine heavy atoms. This value can be compared to 0.046 Å for the overall structure.

Reviewer: 3. The authors should experimentally show that there is a pH dependence on the rate of catalysis in this enzyme.

Answer: *This experiment has been done. We monitored activity versus pH ranging from 5 to 10. Unfortunately, the in vitro reaction turned out to be more complex than only corresponding to L-tyrosine → p-cresol + DHG. Indeed, redox processes to convert p-cresyl• into p-cresol are affected either by pH or by the presence of thiol-containing reducing agents such as DTT or b-mercaptoethanol, leading to alternative outcomes. As a consequence, it is not possible to clearly deconvolute pH effects on either the catalytic rate or on the production of p-cresol versus such alternative outcomes. This feature is complex and out of the scope of the present manuscript as it should deserve a thorough description that would blur the current message.*

Reviewer: 4. Line 25: delete the

Answer: *Done.*

Reviewer: 5. Scheme 1: add space between the top set of reactions and the bottom because they are running together.

Answer: *Done.*

Reviewer: 6. Lines 45-54: HydG and ThiH share 27-percent sequence identity with each other and each share 25-percent sequence identity with NosL? Are these statements interpreted correctly? Line 222 says that ThiH and NosL share less than 23% identity.

Answer: *We thank the Reviewer for pointing this. We have now corrected the values accordingly.*

Reviewer: 7. Add the reaction of HydG, ThiH, and NosL into the supplement.

Answer: *Done. Supplementary Scheme 1 has been added and cited in the main text.*

Reviewer: 8. Add a supplementary figure that shows that comparison being made between BioB, HydE, PylB, HydG, and NosL. The comparison with NosL can be used to back up the statement that the active site of ThiH is reminiscent of NosL.

Answer: *Done. This figure is now the new Supplementary Fig. 1.*

Reviewer: 9. Add a figure to show the described different conformations of chains A and B that includes the crystal contacts.

Answer: *New Supplementary Fig. 2 has been added and is now cited after the discussion on the chain conformational differences.*

Reviewer: 10. In line with the above comment since the region is mentioned to be structurally flexible, include a discussion about the B-factors in this region.

Answer: Flexibility is deduced by the observation of two distinct conformations (New Supplementary Fig. 2). Yet, such flexibility does not mean that these stretches are agitated in the crystal, which is not the case by the way. Therefore, a discussion on the B-factors is not relevant in this case. Besides, this observation matches that we previously observed in HydE, where the C-terminal stretch covers the bottom of the active site cavity at the bottom of the barrel and can be disordered, hence opening the cavity at that position (Rohac et al. JACS 2021). We have also added this reference in the legend of New Supplementary Fig. 2.

Reviewer: 11. For Figure 1d, add a complementary panel that shows the maps of L-Tyr and the interacting residues.

Answer: Done. The map of L-Tyr and the interacting residues has been added as stereoview in a new Supplementary Figure (New Supplementary Fig. 3a) and not in Fig. 1c because the interactions would have been difficult to visualize.

Reviewer: 12. For Figure 1d, add the distances of the highlighted interactions.

Answer: Done. Since it is a scheme and hydrogen atoms are not present in the crystallographic model, distances between the heavy atoms of the protein and the substrate have been added.

Reviewer: 13. For the description of L-Tyr binding in the active site add references to Figure 1d if that is the figure that best shows the described interactions. See lines 136-137 and 137-138.

Answer: Done.

Reviewer: 14. The structures and labels in Figure 2 are too small.

Answer: Panels b and c of Figure 2 have been enlarged accordingly.

Reviewer: 15. Add a figure that supports this statement: "HydG crystal structures display significant structural differences with ThiH".

Answer: New Supplementary Fig. 8 has been added and is now cited after this statement.

Reviewer: 16. With the gained computational insight about the structure-function relationships in the reaction specificity in NosL and ThiH, it would be nice to include an experiment that shows that the reaction outcome can be changed (is there a mutagenesis reaction that could support the claims made?).

Answer: We agree with the reviewer, such experiment would be nice and we already tested several mutations in that prospect. Yet, as stated in our manuscript, while both enzymes display very similar active sites, they only share about 23% sequence identities. Hence, the difference in substrate and reaction specificities comes from the sum of many slight structural differences likely coming from the differences in their overall structures. Therefore, a single mutation cannot convert one into another. This instead would likely require broader protein engineering approaches.

Reviewer: 17. Lines 441-442 state that "this can probably be generalized to radical SAM enzymes given the similar compactness also observed in available 5'dA – substrate bound structures" references just AprD4 and HydE. Is this statement means to be generalized to the entire superfamily of radical SAM enzymes or just this small subset? The former is likely a stretch to base on the three mentioned structures so the statement should be made more clearly.

Answer: We have now modified the sentence to make it clearer. It can indeed be generalized to all radical SAM enzymes. The two references corresponded to the only structures of radical SAM enzymes in complex with both 5'-dA and substrate in addition to the L-tyrosine bound ThiH structure (this work). Yet all the other radical SAM enzyme structures with [SAM]-substrate bound show a similar trend placing the C5' atom near the atom that holds the hydrogen to be abstracted. We have now clarified this point and added two more references.

Reviewer: 18. The abstract mentions that the two enzymes NosL and ThiH share significant primary structure similarities, but the sequence identity seems to be less than 30-percent which does not seem as significant as implied.

Answer: We agree with the reviewer, 30-percent seems low, but is pretty high when comparing with any other radical SAM enzyme family member (usually less than 10-percent). In addition, these 30-percent are located mainly at the active site. In fact, this is because they share these 30-percent identities that we were able to identify L-tyrosine as the substrate of HydG in 2009 (<https://febs.onlinelibrary.wiley.com/doi/full/10.1016/j.febslet.2009.01.004>).

Reviewer: 19. Line 80: "these enzymes work as assembly lines". I am unclear on the meaning of this sentence.

Answer: This has now been clarified. We meant that unlike many other enzymes, in ThiH and HydG substrate accesses the active site on one side of the enzyme, it is processed in the cavity across the protein and product is released on the other side in an unidirectional way like it is usually occurring in factory assembly lines.

REVIEWERS' COMMENTS

Reviewer #1 (Remarks to the Author):

The authors have appropriately addressed all of my questions and I have no further issues with the revised manuscript. I support publication of the revised paper.

Reviewer #2 (Remarks to the Author):

Authors have addressed all my concerns in the revised version of the manuscript.
Accept for publication.

Reviewer #3 (Remarks to the Author):

The reviewers have addressed (all of) the reviewer comments well. I find that the new enlarged figures, subtle rewording of the text, and SI legends lend greatly to the readability of the manuscript.